# The Effects of Gelatinized Wheat Starch and High Salt Diet on Gut Microbiota and Metabolic Disorder

**DOI:** 10.3390/nu12020301

**Published:** 2020-01-22

**Authors:** Moon Ho Do, Hye-Bin Lee, Eunjung Lee, Ho-Young Park

**Affiliations:** 1Research Division of Food Functionality, Korea Food Research Institute, Jeollabuk-do 55365, Korea; Do.moon-ho@kfri.re.kr (M.H.D.); 50023@kfri.re.kr (H.-B.L.); 2Department of Food Science and Technology, Chonbuk National University, Jeollabuk-do 54896, Korea; 3Research Division of Strategic Food Technology, Korea Food Research Institute, Jeollabuk-do 55365, Korea; ejlee@kfri.re.kr

**Keywords:** asian diet, gut microbiota, high-starch diet, metabolic disorder, NAFLD

## Abstract

Diets high in gelatinized starch and high in gelatinized starch supplemented with salt-induced metabolic disorders and changes in gut microbiota have scarcely been studied. In this study, mice on wheat starch diets (WD) exhibited significantly higher body weight, white adipose tissue (WAT), and gut permeability compared to those on normal diet (ND). However, gelatinized wheat starch diet (GWD) and NaCl-supplemented gelatinized wheat starch diet (SGW) mice did not increase body and WAT weights or dyslipidemia, and maintained consistent colon pH at ND levels. WD mice showed higher levels of *Desulfovibrio*, *Faecalibaculum*, and *Lactobacillus* and lower levels of *Muribaculum* compared to ND mice. However, GWD and SGW mice showed a significantly different gut microbial composition, such as a lower proportion of *Lactobacillus* and *Desulfovibrio*, and higher proportion of *Faecalibaculum* and *Muribaculum* compared to WD mice. High starch diet-induced dysbiosis caused increase of lipid accumulation and inflammation-related proteins’ expression, thereby leading to non-alcoholic fatty liver disease. However, GWD and SGW showed lower levels than that, and it might be due to the difference in the gut microbial composition compared to WD. Taken together, diets high in gelatinized starch and high in gelatinized starch supplemented with salt induced mild metabolic disorders compared to native starch.

## 1. Introduction

Asian diets are characterized by high carbohydrate and salt contents, which is reported to be an important cause of metabolic diseases. High-carbohydrate diet (HCD) is known to induce hepatic steatosis and cholesterol deposition in the same manner as a high-fat diet [1]. Starch, which is a major energy source in HCD and the main constituent of wheat flour and polished rice, is a well-known cause of obesity [2]. Starch gelatinization occurs during cooking and involves melting of the crystalline structure, granule swelling, and amylose leaching [3]. Chung et al. reported that gelatinized starch consists of more than 95% rapidly digestible starch [4]. Therefore, gelatinized starch is more bioavailable for amylase hydrolysis than native starch and can cause obesity [5]. Starch is digested and decomposed into glucose. Increased circulating glucose in the body causes inflammation and insulin resistance [6]. Thus, obesity and/or metabolic diseases may be linked to Asian diets, which are typically high in starch.

High salt intake is also known to cause various metabolic diseases. It is the major cause of high blood pressure, leading to cardiovascular diseases [7] and is related to an increased risk of obesity [8]. However, several recent lines of evidence have shown beneficial effects of a high-salt diet, such as reducing hyperglycemia, insulin resistance, and fat deposition [9,10].

It is well known that gut and liver are connected by portal vein and bidirectionally communicate. Changes in the gut microbial composition can cause disruption of the gut barrier function, thereby causing translocation of microbial products, such as endotoxins and microbial-associated molecular patterns [11]. These products are recognized by toll-like receptors (TLRs) and initiate inflammatory cascades [12]. Therefore, metabolic endotoxemia was occurred and induced low-grade inflammation in various tissues, resulting in obesity and non-alcoholic fatty liver disease (NAFLD) [13].

It has been reported that high-carbohydrate or high-salt diets are known to alter gut microbiota. However, changes in gut microbiota caused by diets high in gelatinized wheat starch and high in salt supplemented with gelatinized wheat starch have not yet been reported. Therefore, we investigated the effect of these diets on mice in this study.

## 2. Materials and Methods

### 2.1. Materials

Wheat starch, 4 kDa fluorescein isothiocyanate (FITC)-dextran, and NaCl were obtained from Sigma-Aldrich (St. Louis, MO, USA). Primary antibodies to western blot, such as peroxisome proliferator-activated receptor-γ (PPAR-γ), peroxisome proliferator-activated receptor-α (PPAR-α), sterol regulatory element-binding protein 1 (SREBP-1), acetyl-CoA carboxylase (ACC), fatty acid synthase (FAS), monocyte chemoattractant protein-1 (MCP-1), interleukin-1β (IL-1β), interleukin-6 (IL-6), TLR-4, tumor necrosis factor-α (TNF-α), and β-actin, were purchased from Abcam (Cambridge, MA, USA). Mouse and rabbit secondary antibodies were purchased from Thermo Fisher Scientific (Waltham, MA, USA). Tissue-Tek^®^ O.C.T.^™^ Compound was purchased from Sakura (Alphen aan den Rijn, Netherlands)

### 2.2. Preparation of Gelatinized Wheat Starch

To prepare gelatinized wheat starch, 10 kg of wheat starch powder was heated and stirred in 100 L of water for 60 min at 70 °C, quickly cooled to −70 °C, and lyophilized. Lyophilized gelatinized starch was used for mouse diets.

### 2.3. Animals and Diets

Six-week-old male C57BL/6J mice were purchased from Central Lab Animal Inc. (Seoul, Korea) and the animals were maintained in a humidity-controlled animal facility with a 12-h light/dark cycle at 23 °C and provided with food and water ad libitum. All animal studies were approved by the Animal Welfare Committee of the Korea Food Research Institute (KFRI-M-18025). After acclimation, mice were divided into 4 groups (*n* = 9) and fed a normal diet (ND), high-wheat starch diet (WD), gelatinized wheat starch diet (GWD), and NaCl-supplemented gelatinized wheat starch diet (SGW) for 8 weeks. Mice were housed three per cage and the body weight was recorded every week and food intake was measured two times a week until the end of the study.

The 2018S Teklad Global (Harlan, Madison, WI, USA) was used for the ND diet and its composition was 18.0% of calories in fat, 24.0% of calories in protein, and 58.0% of calories in carbohydrate. WD contained 17.1% of calories as fat (8.8% from soybean oil and 91.2% from lard), 17.3% of calories as protein (97.5% from casein and 2.5% from L-cysteine), and 65.6% of calories as carbohydrates (85% from wheat starch and 15% from sucrose). The composition of GWD was the same as that of WD except that wheat starch was replaced by gelatinized wheat starch. SGW was composed of 4% NaCl in addition to GWD (Table 1).

### 2.4. Blood Serum Analysis

After 8 weeks, the mice were fasted for 12 h and euthanized by anesthesia. Blood was drawn into microfuge tubes, and serum was collected thereafter. Total cholesterol and LDL cholesterol levels were quantified using a Cholesterol Assay Kit (Abcam, Cambridge, MA, USA), and triglyceride levels were analyzed using a Triglyceride Assay Kit (Abcam, Cambridge, MA, USA), according to the manufacturers’ instructions.

### 2.5. Western Blotting

Total protein was extracted from the liver with PRO-PREP™ (iNtRON Biotechnology, Seongnam, Korea). Proteins were separated using a polyacrylamide gel and transferred onto a polyvinylidene difluoride membrane. The membranes were incubated overnight with the primary antibodies against PPAR-γ, PPAR-α, SREBP-1, ACC, FAS, MCP-1, IL-1β, IL-6, TLR-4, TNF-α, and β-actin at 4 °C and then incubated with secondary antibodies at 23 °C. Then, bands were detected using a ChemiDoc™ XRS+ imaging system (Bio-Rad, Hercules, CA, USA).

### 2.6. Gut Microbiota Analysis

At week 4 and 8, fresh fecal samples were collected, and fecal DNA was extracted for the analysis of microbial composition. Hypervariable V3–V4 region of 16S rRNA amplicons were generated using a MiSeq (Illumina, San Diego, CA, USA) at Macrogen (Seoul, Korea) according to the manufacturer’s instructions. Microbial diversity was assessed using the QIIME software and operational taxonomic units (OTUs) were defined at ≥97% sequence homology [14]. Taxonomic composition was generated using QIIME-UCLUST based on the Ribosomal Database Project [15]. Principal coordinate analysis plots (PCoA) were generated at the genus level based on Bray-Curtis dissimilarity [16].

### 2.7. Intestinal Permeability and Colon pH Analysis

Intestinal permeability was analyzed at 8 weeks as previously described [17]. Mice were fasted briefly for 6 h and received 500 mg/kg of FITC-dextran. After 2 and 4 h, blood was collected from the tail vein and plasma was obtained. The intensity of fluorescence was measured using a microplate reader (Molecular Devices, Sunnyvale, CA, USA) at 485/535 nm excitation/emission wavelength. Standard curve was prepared by diluted FITC-dextran in non-treated plasma. Colon pH was measured immediately after euthanization on week 8. Colon contents were collected and mixed with deionized water. Then, the pH value was measured using a pH meter (Orion Star A211, Thermo Scientific, Waltham, MA, USA).

### 2.8. Histological Analysis

Hematoxylin and eosin (H&E) staining and oil red O (ORO) staining were performed for histological analysis. For H&E staining, liver and WAT were fixed with 4% formaldehyde and processed with paraffin. Then, the tissues were sliced and stained with H&E. For ORO staining, liver tissues were frozen with O.C.T. Compound and sliced. Then, sections were fixed in 4% paraformaldehyde and then stained with ORO. All sections were scanned and analyzed using a Pannoramic 250 Flash III slide scanner (3DHISTECH Ltd., Budapest, Hungary) and CaseViewer software (3DHISTECH Ltd).

### 2.9. Statistical Analysis

The data are presented as the mean ± standard error of the mean. One-way ANOVA analysis and Tukey’s analysis were used for the statistical significance of the differences among groups. All data were obtained using GraphPad Prism software (San Diego, CA, USA). A *p*-value <0.05 was considered statistically significant.

## 3. Results

### 3.1. Effects of Diet on Body Weight and Metabolic Parameters

Over the course of 8 weeks, WD mice showed a significant increase in body weight (Figure 1a) and their epididymal WAT weight was two-fold higher (Figure 1c) than that of ND mice. However, GWD and SGW mice did not show an increase in body and WAT weight without a change of food intake (Figure 1b). H&E staining demonstrated that the WAT size in WD mice increased significantly while WAT remained the same size in GWD, SGW, and ND mice. To confirm diet-induced dyslipidemia, total cholesterol, LDL cholesterol, and triglyceride levels were determined. WD mice showed a significant increase in total cholesterol and LDL cholesterol levels, but GWD and SGW mice showed similar levels compared to ND (Figure 1d,e). Triglyceride levels did not differ between ND and WD mice; however, GWD and SGW mice showed a significant decrease (Figure 1f). These results suggest that HCD induces metabolic disorder; however, the diet with high salt supplementation and gelatinized starch did not affect obesity.

### 3.2. Effects of Diet on Gut Microbial Diversity and Composition

All HCD groups showed fewer OTUs and lower Shannon and Simpson indices compared to the ND group (Figure 2a). Principal coordinate analysis indicated obvious separation between ND and WD groups (Figure 2b). GWD and SGW groups were located closely but clearly separated from ND and WD groups. The significant changes in the gut microbial composition of each group were displayed in taxon-based analysis (Figure 2c). At the phylum level, WD mice showed significantly higher proportion of Firmicutes and Proteobacteria and significantly lower proportion of Bacteroidetes. However, GWD and SGW not only reversed this pattern but also showed higher levels of Verrucomicrobia. In particular, as shown in Figure 2d, WD mice showed higher levels of *Desulfovibrio*, *Faecalibaculum*, *Romboutsia*, and *Lactobacillus* genera and lower levels of *Alistipes* and *Muribaculum* genera compared to ND mice. Compared to WD mice, GWD mice have a lower proportion of *Lactobacillus* and *Desulfovibrio* and higher proportion of *Faecalibaculum*, *Muribaculum*, and Bacteroides. Interestingly, the gut microbial composition of SGW was similar to GWD mice. Moreover, GWD and SGW showed a significantly higher proportion of *Akkermansia*, which has been reported to combat obesity. In summary, high-starch diet reduced the gut microbial diversity and markedly changed the gut microbial composition; however, diets high in gelatinized starch and high salt supplementation shifted the gut microbial composition.

### 3.3. Effects of Diet on Bowel Health

In gut permeability analysis, the levels of plasma FITC-dextran and area under the curve (AUC) significantly increased in every HCD group. Among these, SGW mice exhibited the highest increase in permeability (Figure 3a,b). These results suggest that HCD and the diet with high salt supplementation and gelatinized starch disrupt the intestinal barrier function. However, colon pH exhibited a different pattern than gut permeability. As shown in Figure 3c, WD mice had significantly higher colon pH, but GWD and SGW mice had the same pH as ND mice. These results may be due to differences in metabolites produced by diet-induced alteration of gut microbiota.

### 3.4. Effects of Diet on Liver Lipid Metabolism

To evaluate diet-induced changes in lipid metabolism, protein expression of lipid metabolism regulatory factors, such as SREBP-1, PPAR-α, and PPAR-γ, was analyzed. SREBP-1 and PPAR-γ protein expression increased two-fold while PPAR-α protein expression decreased in the WD mice liver (Figure 4). Moreover, upregulated SREBP-1 and PPAR-γ expression led to a significant increase in transcription factors known to play an important role in lipid accumulation, such as ACC and FAS. GWD and SGW mice showed markedly lower levels of SREBP-1, PPAR-γ, ACC, and FAS and higher levels of PPAR-α compared to WD mice. However, GWD and SGW protein expression was different when compared to ND mice. These results indicated that the diet high in gelatinized starch may induce mild lipid accumulation in liver than HCD and the diet with high salt supplementation in gelatinized starch did not affect lipid metabolism.

### 3.5. Effects of Diet on Liver Inflammation

The correlation of changes in the gut microbiota with diet-induced metabolic disease was assessed by comparing inflammatory cytokine expression. As shown in Figure 5, MCP-1, IL-1β, IL-6, TLR-4, and TNF-α protein expression in the liver was significantly higher in WD compared to ND mice. Interestingly, GWD mice showed a significant decrease in inflammatory factor expression than WD. The SGW group also exhibited lower levels of pro-inflammatory cytokines compared to WD but slightly higher than GWD. These results suggest that starch can induce metabolic disorder through liver inflammation, but gelatinized starch did not affect liver inflammation. However, high salt intake may negatively affect the liver directly or indirectly. Further studies are needed to determine the effect of high salt intake on liver in HCD.

### 3.6. Histological Changes

Development of NAFLD in the liver was detected using H&E staining and ORO staining. As shown in Figure 6, WD mice exhibited severe hepatic fat deposition in contrast to GWD and SGW mice, which exhibited only mild fat deposition. ORO staining also showed that HCD can lead to the development of NAFLD and can be alleviated by substitution with gelatinized starch. This data indicates that the intake of gelatinized starch may induce mild NAFLD than HCD and that high salt intake did not make NAFLD worse.

## 4. Discussion

High carbohydrate and/or high salt diets are known to be one of the major causes of metabolic diseases. Gelatinized starch is also considered to be a potent inducer of obesity than native starch. Starch gelatinization occurs by granule swelling and crystalline structure loss in the presence of heat and water. It involves breaking of intermolecular hydrogen bonds into amylose and amylopectin, thus increasing starch digestibility. Starch gelatinization is one of the most important factors in the determination of digestibility and nutritional properties. It is well known that native starch is degraded slowly by amylases but gelatinization of starch increases sensitivity to amylases [18]. Therefore, gelatinized starch is rapidly hydrolyzed to saccharides, and glucose and insulin responses than native starch [19]. However, its effects on metabolic diseases and change of gut microbiota are not yet studied. For these reasons, we conducted this study to confirm the effect of diets with high gelatinized starch and salt supplementation. High-starch diets are known to cause obesity and metabolic diseases [20]. Our results showed that HCD led to a significant increase in body weight and cholesterol levels, but diets high in gelatinized starch, and salt supplementation on gelatinized starch did not lead to an increase in body weight and resulted in stable cholesterol levels. In addition, triglyceride levels in GWD and SGW groups were found to be lower than ND. These results are contrary to previous research that gelatinized starch and high salt diets induce obesity [21,22]. However, Ramirez showed that high levels of gelatinized starch suppressed growth rate than low-gelatinized starch and the type of starch might be an important determinant of obesity [23]. Moreover, mice that are unable to metabolize fructose protected against metabolic disorders [22]. Taken together, we assumed that without body weight change and low triglyceride levels in the GWD and SGW groups cannot metabolize gelatinized starch induced by the change of the carbohydrate structure.

To determine whether the changes in body weight and lipid metabolism were caused by changes in gut microbial composition, 16s rRNA analysis was performed. The high-starch diet exhibited significantly increased Fermicutes to Bacteroidetes ratio and increased proportion of Proteobacteria compared to ND. This is consistent with the gut microbial composition in metabolic diseases revealed by many researchers [24]. The *Muribaculum* genus, previously classified as S24-7, is known to decrease in mice on high-fat diet and is associated with the regulation of body weight [25,26]. Moreover, Romboutsia, which is an obesity-related phylotype, is positively associated with lipid profile and lipogenesis in the liver [27]. In this study, WD showed a significantly reduced proportion of Muribaculum and increased proportion of Romboutsia, which may have led to weight gain. It has been reported that *Lactobacillus* spp. treatment has an anti-obesity and probiotic effect in diet-induced obesity [28]. However, our results showed a great increase in *Lactobacillus* in WD, which is expected to be strongly associated with obesity or metabolic disease. In support of our results, several studies have reported a significant increase in the proportion of *Lactobacillus* in NAFLD [29]. *Faecalibaculum*, one of lactic acid producers, has not been studied as much in this context yet, but its role in metabolic disease has been studied to some extent [30]. These changes in gut microbial composition caused by WD are thought to lead to an increase in body weight and change in lipid metabolism. Many studies have reported that resistant starch or high-salt intake may induce alteration of gut microbial composition and metabolic diseases [31,32]. However, no studies have reported changes in gut microbial composition in diet high in salt supplemented with gelatinized starch. Our results showed that consumption of gelatinized starch leads to a gut microbial composition that is significantly different from that of native starch. The GWD group showed a higher proportion of *Muribaculum* compared to WD mice, and the proportion of *Lactobacillus* was almost similar to ND mice. Interestingly, the SGW group had a similar gut microbial composition to GWD and it also showed a markedly higher composition of *Muribaculum* and lower composition of *Lactobacillus* than those of WD. It can be concluded that the change in gut microbiota by diets high in gelatinized starch and high in salt supplemented with gelatinized starch cause weight and lipid metabolism to be similar to ND mice. In addition, the increased proportion of *Akkermansia*, known to be an intestinal microorganism with anti-obesity effects, was confirmed. This also seems to have played a critical role in maintaining normal weight in GWD and SGW mice, despite having a lower proportion of *Muribaculum* compared to ND mice. Therefore, both groups showed a higher proportion of *Faecalibaculum* than WD mice, which is expected to have a negative impact on other tissues.

HCD is known to cause NAFLD by inducing inflammation and altering lipid metabolism in the liver [33]. The changes in lipid metabolism may be due to changes in gut microbiota-induced alteration of the gut–liver axis [34]. Alterations in the gut microbiota and increase in gut permeability enhance the exposure of gut-derived microbial products to the liver, thereby inducing endotoxemia and gut–liver axis alteration. [35]. Our results showed that HCD changed the lipid metabolism by leading to increased PPAR-γ and SREBP-1 expression and decreased PPAR-α expression. SREBP-1 and PPAR-γ are known to accelerate adipogenesis [36]. PPAR-α, on the other hand, is known to regulate lipid metabolism by increasing fatty acid oxidation [37]. Several lines of evidence have shown that SREBP-1 and PPAR-γ are key regulators of FAS and ACC [38,39]. FAS and ACC were also significantly increased in WD mice, suggesting that HCD can cause NAFLD through alteration of gut–liver axis-induced lipid accumulation. The GWD group also showed changes in lipid metabolism-related protein expression, but it was milder than WD. SGW-fed mice also exhibited significantly lower levels of lipid metabolism-related proteins expression compared to WD and, interestingly, FAS expression was more downregulated than GWD. These results may due to salt supplementation contributing to the regulation of lipid metabolism by GWD. It is consistent with a previous study that high salt intake changes lipid metabolism, such as decreased lipogenesis and increased lipolysis [10]. Taken together, we suggest that GWD and SGW can induce NAFLD through changed gut microbial composition. Histological analysis by H&E and ORO staining demonstrated that WD mice showed increased fat deposition, but the GWD, SGW, and ND groups exhibited relatively poor fat deposition. These results are consistent with changes in the expression of lipid accumulation-related factors and indicate that NAFLD can be induced in GWD and SGW mice, even though gelatinized starch did not induce weight gain. However, higher levels of *Muribaculum* and *Akkermansia* and lower levels of *Lactobacillus* may induce a decrease in gut–liver axis alteration and lipid metabolism, resulting in mild NAFLD.

It is well known that endotoxins, produced by gut microbiota, induce TLR-4 activation through alteration of the gut–liver axis, thereby inducing a pro-inflammatory response by TNF-α and IL-6 activation in the liver [40]. Therefore, overexpression of TLR-4 plays an important role in the development of NAFLD. Moreover, IL-1β production involved in NAFLD could suppress expression of PPAR-α and indirectly induce TNF-α-induced cell death [41]. Our results also showed that HCD significantly increased the expression of inflammatory cytokines compared to ND, but interestingly, GWD did not increase the expression of these proteins. These results suggested that changing carbohydrate sources from native starch to starch with altered structure like gelatinization may help alleviate the fat accumulation and NAFLD. Some studies reported a pro-inflammatory role with an overcolonization of *Akkermansia* [42]. However, in spite of the increased proportion of *Faecalibaculum* and decreased proportion of *Muribaculum*, GWD did not showed an increase of inflammatory cytokines. These results indicated that an increased proportion of *Akkermansia* may play a positive role in liver inflammation. Although the SGW group showed lower expression of inflammatory cytokines than WD, on the other hand, they exhibited a significant increase in the expression of inflammatory cytokines compared to GWD mice. However, high salt supplementation did not increase TLR-4 expression, so it may not be due to changes in gut microbiota-induced alteration of the gut–liver axis. Increased inflammatory cytokines without elevation of TLR-4 suggest that high salt supplementation has a direct negative effect on the liver. These results may be related to higher gut permeability in SGW mice. Further research is needed, however, to determine the effects of high-salt intake on the gut–liver axis and liver inflammation.

## 5. Conclusions

GWD and SGW altered WD-induced change of gut microbiota, such as an increase of Bacteroidetes and decrease of Firmicutes and Proteobacteria. Hence, it can be concluded that gelatinized starch did not increase body and WAT weight. Moreover, high salt supplementation did not affect gelatinized starch-induced change in gut microbiota. Both GWD and SGW groups had milder NAFLD compared to WD. These results may be due to decreased *Muribaculum* and increased *Faecalibaculum* compared to ND. For these reasons, we carefully suggest that diets high in gelatinized starch and high in gelatinized starch supplemented with salt cause mild metabolic disorders compared to native starch. However, further studies on the increased expression of inflammatory cytokines in liver and gut permeability in high salt-supplemented diets are needed.

## Figures and Tables

**Figure 1 nutrients-12-00301-f001:**
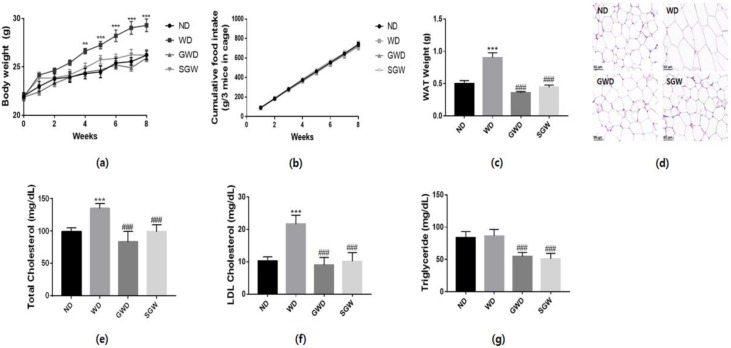
Changes of body and white adipose tissue (WAT) weight and lipid metabolism. (**a**) Body weight changes during the 8 weeks of feeding; (**b**) Cumulative food intake during the 8 weeks of feeding; (**c**) WAT weight; (**d**) Hematoxylin and eosin (H&E) staining of WAT; (**e**) Total cholesterol levels; (**f**) Low-density lipoprotein cholesterol levels; (**g**) Triglyceride levels. Data are presented as the mean ± SEM for nine mice per each group (** *p* < 0.01, *** *p* < 0.001 vs. ND; ### *p* < 0.01 vs. WD).

**Figure 2 nutrients-12-00301-f002:**
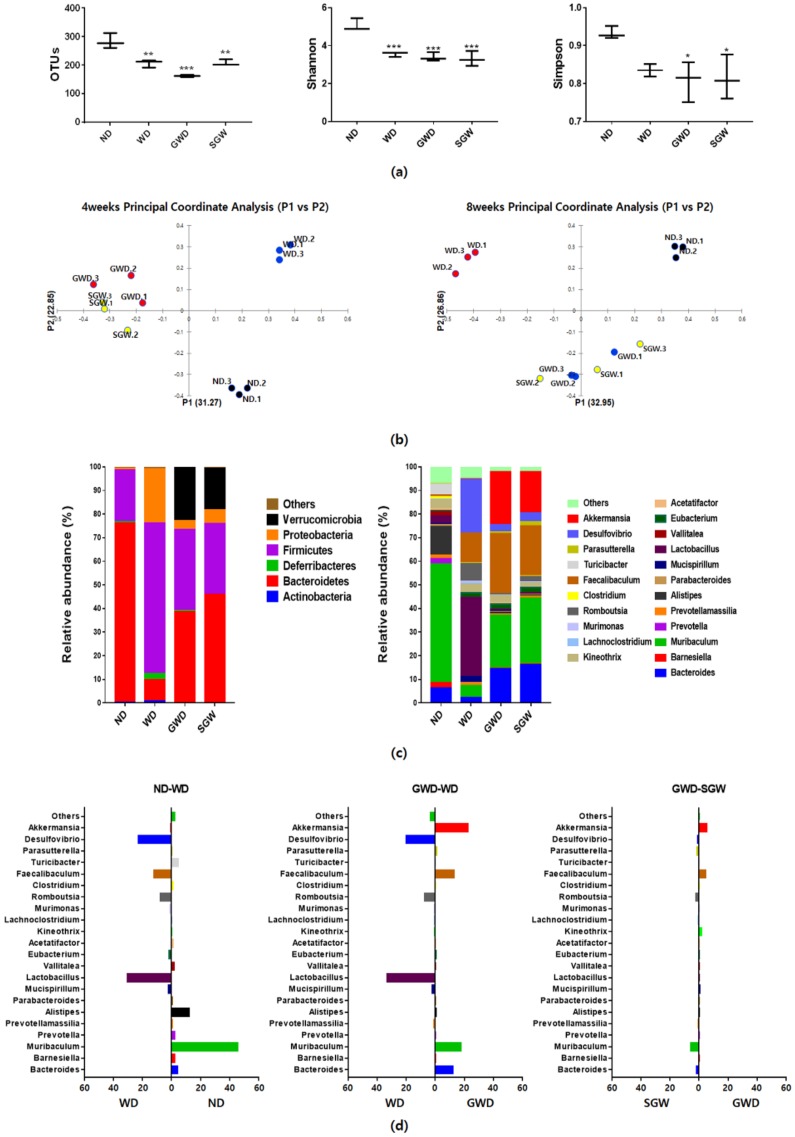
Microbial community analysis of feces samples. (**a**) Operational taxonomic unit levels, Shannon’s and Simpson diversity indices; (**b**) Bray–Curtis dissimilarity-based principal coordinates analysis; (**c**) Relative abundances plot of bacterial phyla and genus; (**d**) Different abundance of gut microbial composition by diet. Data are presented as the mean ± SEM for three cages per each group (* *p* < 0.05, ** *p* < 0.01, *** *p* < 0.001 vs. ND).

**Figure 3 nutrients-12-00301-f003:**
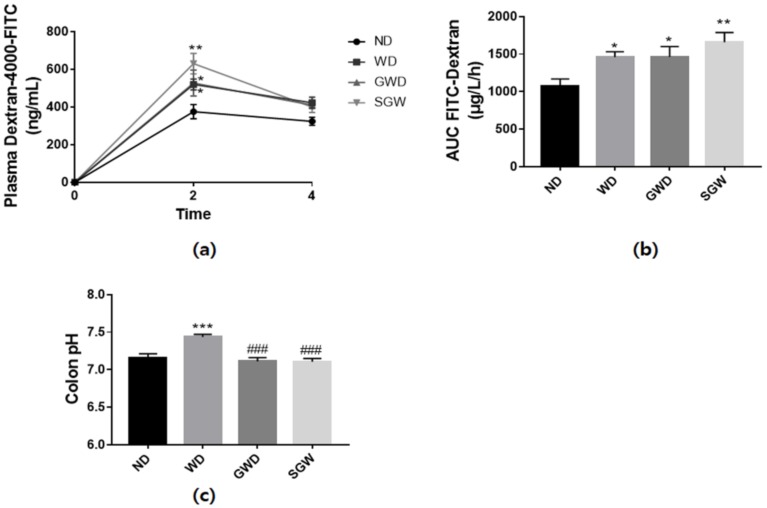
Diet-induced changes of gut barrier function. (**a**) Plasma fluorescein isothiocyanate (FITC)-dextran concentration during the gut permeability test; (**b**) Area under the curve (AUC) of plasma FITC dextran levels; (**c**) Colon pH levels. Data are presented as mean ± SEM for 9 mice per each group. (* *p* < 0.05, ** *p* < 0.01, *** *p* < 0.001 vs. ND; ### *p* < 0.001 vs. WD).

**Figure 4 nutrients-12-00301-f004:**
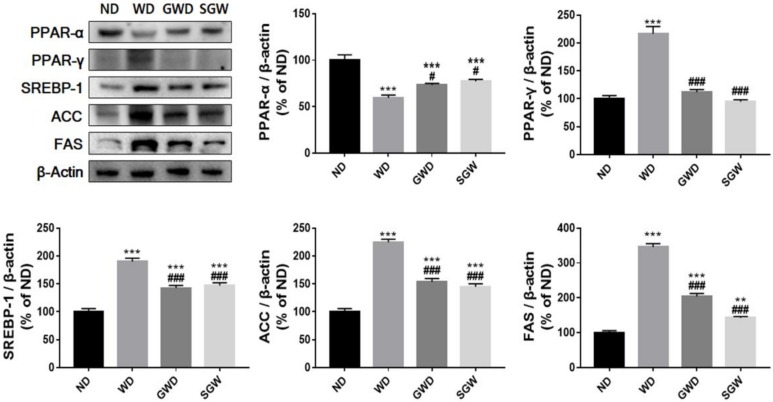
Diet-induced change of lipid metabolism in liver. Band intensities were normalized to those of β-actin and bar values are presented as the mean percentage of ND ± SEM of three independent experiments (** *p* < 0.01, *** *p* < 0.001 vs. ND; # *p* < 0.05 and ### *p* < 0.001 vs. WD).

**Figure 5 nutrients-12-00301-f005:**
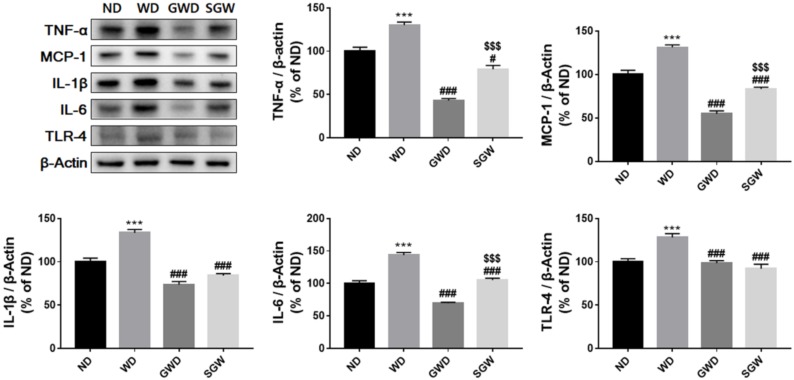
Diet-induced change of inflammatory cytokines’ expression in the liver. Band intensities were normalized to those of β-actin and bar values are presented as the mean percentage of ND ± SEM of three independent experiments (* *p* < 0.05, *** *p* < 0.001 vs. ND; # *p* < 0.05, ### *p* < 0.001 vs. WD; $$$ *p* < 0.001 vs. GWD).

**Figure 6 nutrients-12-00301-f006:**
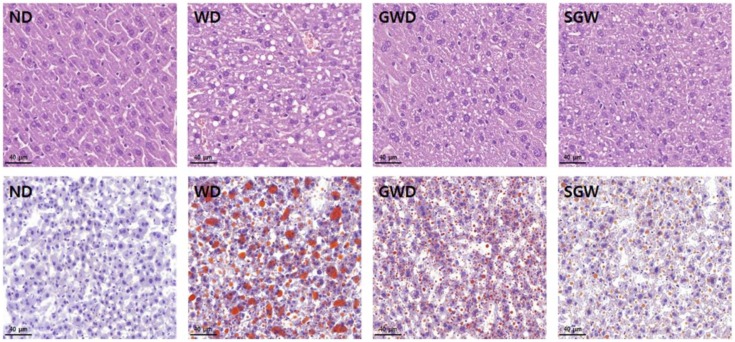
Diet-induced histological changes in liver. Representative histological results by hematoxylin and eosin (upper) and oil red O (lower) staining.

**Table 1 nutrients-12-00301-t001:** Diet composition.

Constituents (g/kg)		WD	GWD	SGW
Protein	Casein	157.3	157.3	157.3
	L-cystine	4.0	4.0	4.0
Carbohydrate	Sucrose	90.0	90.0	90.0
	Wheat starch	550.0	-	-
	Gelatinized wheat starch	-	550.0	550.0
Fat	Lard	54.0	54.0	54.0
	Soybean oil	5.2	5.2	5.2
Extra	Cellulose	65.5	65.5	65.5
	Vitamin mix, AIN76A	10.0	10.0	10.0
	Mineral mix, AIN76	62.0	62.0	22.5
	Choline bitartrate	2.0	2.0	2.0
	Sodium chloride	-	-	39.5
	Ethoxyquin	0.01	0.01	0.01
**Constituents (% by weight)**				
Protein		14.1	14.1	14.1
Carbohydrate		51.9	51.9	51.9
Fat		6.4	6.4	6.4
Total kcal		321.2	321.2	321.2
**Constituents (%kcal from)**				
Protein		17.5	17.5	17.5
Carbohydrate		64.6	64.6	64.6
Fat		17.8	17.8	17.8
**Energy density (kcal/g)**		3.2	3.2	3.2

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
