# Peer review of "The Effects of Gelatinized Wheat Starch and High Salt Diet on Gut Microbiota and Metabolic Disorder"

_nutrients, 2020, doi:10.3390/nu12020301_

Round 1
Reviewer 1 Report
The questions were answered. However, I still think the summary needs to be improved.
Moreover, I see the need to add to the discussion the contribution of salt (when coupled with gelatinized wheat starch in caused mild metabolic disorders compared native starch. Physiologically, what happens in this process?
L. 319: "Although GWD gut microbial composition is different from ND mice, GWD had higher levels of Akkermansia than WD mice". This sentence should not be in the conclusion.
Author Response
We highly appreciate the reviewer’s constructive and helpful comments on our manuscript. As suggested by the reviewer, we have carefully response (marked in blue) to address the reviewer’s comments and revised manuscript (marked in red). We hope that the reviewer will find our responses to the comments satisfactory.
Reviewer(s)' Comments to Author:
Reviewer 1.
Moreover, I see the need to add to the discussion the contribution of salt (when coupled with gelatinized wheat starch) in caused mild metabolic disorders compared native starch. Physiologically, what happens in this process?
â–¶ According to the reviewer’s comments, we added text as below;
Line 269-273: GWD group showed a higher proportion of Muribaculum compared to WD mice, and the proportion of Lactobacillus was almost similar to ND mice. Interestingly, SGW group had similar gut microbial composition to GWD and it also showed markedly higher composition of Muribaculum and lower composition of Lactobacillus than those of WD. Line 291-296: GWD group also showed change in lipid metabolism-related protein expression, but it was milder than WD. SGW fed mice also exhibited significantly lower levels of lipid metabolism related proteins expression compared to WD and, interestingly, FAS expression was more down-regulated than GWD. These result may due to salt supplementation contributing to the regulation of lipid metabolism by GWD. It is consistent with previous study that high salt intake changes lipid metabolism such as decreased lipogenesis and increased lipolysis [10]. Line 316-318: Although SGW group showed lower expression of inflammatory cytokines than WD, on the other hand, exhibited a significant increase in the expression of inflammatory cytokines compared to GWD mice.L 319: "Although GWD gut microbial composition is different from ND mice, GWD had higher levels of Akkermansia than WD mice". This sentence should not be in the conclusion.
â–¶ According to the reviewer’s comments, we revised manuscript.
Line 325-326: GWD and SGW altered WD-induced change of gut microbiota such as increase of Bacteroidetes and decrease of Firmicutes and Proteobacteria.

Reviewer 2 Report
In the revised manuscript by Ho Do et al, the authors have attempted to address the reviewer’s comments, but their revisions still require work and do not adequately address the concerns.
1) One of my major comments was that the authors were lacking a WD w/ salt group as a control to the gelatinized wheat w/ salt group. The authors did not add this group, and did not mention or address this in any of their revised text. This is something that must be addressed, because is their changes from addition of salt due just to salt or an interaction only with salt and gelatinized wheat – this is completely unknown with the current groups. Additionally, the title is misleading because of this lack of group – you are not looking at the effect of gelatinization of wheat starch AND high salt… it is WITH high salt, a very big difference.
2) Another major concern was whether the effects of decreased body weight and adiposity are due to decreased food intake – the authors address this by adding the food intake data, but as presented it is almost impossible to decipher. It would be better if, similar to body weight, it was cumulative food intake over time – that would be easier to compare overall total food intake to that of overall body weight gain. Also, there is no information on housing of the mice, are they individually housed? Otherwise how they measured the food intake is not accurate in the methods and not accurate how presented (g/day/mouse).
3) Another major comment was their discussion on previous literature regarding the effects of gelatinization on body weight/adiposity. I highlighted the fact that in their original submission, the paper they cited (Diet texture, moisture and starch type in dietary obesity, 1987) actually supported their findings of gelatinized starch diet decreasing adiposity (although the original submission said the 1987 paper showed an increase in obesity). The authors reconciled this by completely removing the text and the citation from the revised manuscript. This is unacceptable, and again, they need to fully address this in their discussion. In fact, the authors now state in the discussion (line 243): “These results are contrary to previous research that gelatinized starch and high salt diets induce obesity [21,22]”. When researching these citations, neither show gelatinized starch induce obesity, and again, the previous paper from 1987 is no longer cited.
4) There are no n’s for their western blot data – they say percentage of 3 independent experiments, but is that an n=1 for each group in each experiment?
5) The revised text is very difficult to understand at times, and has many grammatical errors. For example: Line 245-247: “However, Miguel et al. showed that unable to metabolized fructose mice did not increase triglyceride accumulation [22]. For this reason, we assumed that the low triglyceride levels in GWD and SGW groups are due to unable to metabolize gelatinized starch induced by change of carbohydrate structure.”
Author Response
We highly appreciate the reviewer’s constructive and helpful comments on our manuscript. As suggested by the reviewer, we have carefully response (marked in blue) to address the reviewer’s comments and revised manuscript (marked in red). We hope that the reviewer will find our responses to the comments satisfactory.
Reviewer(s)' Comments to Author:
Reviewer 2.
1) One of my major comments was that the authors were lacking a WD w/ salt group as a control to the gelatinized wheat w/ salt group. The authors did not add this group, and did not mention or address this in any of their revised text. This is something that must be addressed, because is their changes from addition of salt due just to salt or an interaction only with salt and gelatinized wheat – this is completely unknown with the current groups. Additionally, the title is misleading because of this lack of group – you are not looking at the effect of gelatinization of wheat starch AND high salt… it is WITH high salt, a very big difference.
â–¶ The dietary characteristics of Asian people are high carbohydrate diet and high salt diet. Peoples are consumed carbohydrate source by cooking, so they consume it in the form of gelatinized starch rather than native starch. Our hypothesis is that gelatinized starch will lead to severe metabolic disorders than native starch, and intake of gelatinized starch with high salt will exacerbate the metabolic disorder. Therefore, we only performed experiments with WD, GWD and SGW, not with WD + salt. We will confirm the further study for effects of high salt on native starch.
â–¶ According to reviewer’s comments, we revised title
“The effects of gelatinized wheat starch and high salt on gut microbiota and metabolic disorder”
2) Another major concern was whether the effects of decreased body weight and adiposity are due to decreased food intake – the authors address this by adding the food intake data, but as presented it is almost impossible to decipher. It would be better if, similar to body weight, it was cumulative food intake over time – that would be easier to compare overall total food intake to that of overall body weight gain. Also, there is no information on housing of the mice, are they individually housed? Otherwise how they measured the food intake is not accurate in the methods and not accurate how presented (g/day/mouse).
â–¶ According to reviewer’s comments, we revised manuscript and figure 1.
Line 78-79: Mice were housed three per cage and body weight was recorded every week and food intake was measured two times a week until the end of the study.
Line 146: (b) Cumulative food intake during the 8 weeks of feeding;
3) Another major comment was their discussion on previous literature regarding the effects of gelatinization on body weight/adiposity. I highlighted the fact that in their original submission, the paper they cited (Diet texture, moisture and starch type in dietary obesity, 1987) actually supported their findings of gelatinized starch diet decreasing adiposity (although the original submission said the 1987 paper showed an increase in obesity). The authors reconciled this by completely removing the text and the citation from the revised manuscript. This is unacceptable, and again, they need to fully address this in their discussion. In fact, the authors now state in the discussion (line 243): “These results are contrary to previous research that gelatinized starch and high salt diets induce obesity [21,22]”. When researching these citations, neither show gelatinized starch induce obesity, and again, the previous paper from 1987 is no longer cited.
â–¶ According to the reviewer’s comments, we revised manuscript.
Line 244-245: However, Ramirez showed that high level of gelatinized starch suppressed growth rate than low-gelatinized starch and the type of starch might be an important determinant of obesity. [23].[23] Ramirez, I. Diet texture, moisture and starch type in dietary obesity. Physiology & behavior 1987, 41, 149-154.
4) There are no n’s for their western blot data – they say percentage of 3 independent experiments, but is that an n=1 for each group in each experiment?
â–¶ Tissue samples of 6 mice were pooled and performed 3 independent experiments.
5) The revised text is very difficult to understand at times, and has many grammatical errors. For example: Line 245-247: “However, Miguel et al. showed that unable to metabolized fructose mice did not increase triglyceride accumulation [22]. For this reason, we assumed that the low triglyceride levels in GWD and SGW groups are due to unable to metabolize gelatinized starch induced by change of carbohydrate structure.”
â–¶ According to reviewer’s comments, we revised manuscript.
Line 245-248: Moreover, mice that unable to metabolize fructose protected against metabolic disorders [22]. Taken together, we assumed that without body weight change and low triglyceride level in GWD and SGW groups are due to cannot metabolize gelatinized starch induced by change of carbohydrate structure.

Reviewer 3 Report
My comments were addressed appropriately in most of the cases. There are still issues with the language in the abstract.
L3. supplementation can be deleted
L15-17. The sentence doesn’t seem complete. Can be revised to : “ In this study, mice on wheat starch diets (WD) exhibited significantly higher body weight, white adipose tissue (WAT) and gut permeability compared to those on normal diet (ND)”
The language of other part of the abstract must be improved.
Author Response
We highly appreciate the reviewer’s constructive and helpful comments on our manuscript. As suggested by the reviewer, we have carefully response (marked in blue) to address the reviewer’s comments and revised manuscript (marked in red). We hope that the reviewer will find our responses to the comments satisfactory.
Reviewer(s)' Comments to Author:
Reviewer 3.
My comments were addressed appropriately in most of the cases. There are still issues with the language in the abstract.
L3. supplementation can be deleted
â–¶ According to the reviewer’s comment, we revised the title as below;
“The effects of gelatinized wheat starch and high salt on gut microbiota and metabolic disorder”
L15-17. The sentence doesn’t seem complete. Can be revised to : “ In this study, mice on wheat starch diets (WD) exhibited significantly higher body weight, white adipose tissue (WAT) and gut permeability compared to those on normal diet (ND)”
â–¶ According to the reviewer’s comments, we corrected the sentence.
The language of other part of the abstract must be improved.
â–¶ According to the reviewer’s comments, we improved the language of the abstract.
